# Development and Forecast of Employment in Forestry in the Czech Republic

**Daniel Toth [1],\*, Mansoor Maitah [2]**  **and Kamil Maitah [3]**

[1] Department of Forestry and Wood Economics, Faculty of Forestry and Wood Sciences of the Czech University of Life Sciences in Prague, Kamýcká 129, 16500 Prague, Czech Republic

[2] Department of Economics, Faculty of Economics and Management, Czech University of Life Sciences in Prague, Kamýcká 129, 16500 Prague, Czech Republic; maitah@pef.czu.cz

[3] Department of Trade and Finance, Faculty of Economics and Management, Czech University of Life Sciences in Prague, Kamýcká 129, 16500 Prague, Czech Republic; maitahk@pef.czu.cz

\* Correspondence: tothd@pef.czu.cz; Tel.: +420-606-810-534

**Abstract:** Employment in forestry is an essential component of the forestry industry. It is a socio-economic phenomenon, which has been at the edge of economists' interest for quite a long time. The proportion of employees in the forestry sector is relatively small, standing at only 0.6%. However, forestry as a sector has a very significant multiplier effect which is reflected in the growth of related jobs. Examples of this can be found in the production of forestry machinery and equipment, the construction of wooden and timber structures, and the furniture sector. These sectors are kept separately in economic and statistical records, but forestry remains their natural determinant. The aim of this work is to describe, analyze, and formulate the prognosis for the development of these types of jobs. Conclusions of the work show that there has been a decrease in employment and simultaneously an increase in labor productivity. This is due to a increasingly high use of technological equipment. Development forecasts show that the Czech Republic does not differ from the overall surveyed trends in other selected countries. It is therefore evident that forecasts of the development of employment in forestry are also relevant in other similar countries. Our results show a statistically significant reduction in forestry employment. The analysis focused on the Czech Republic, but the results may also apply to other European countries. A significant decrease in employment leads to instability in the forestry sector. It means a skilled labor force leaves the forestry sector and is not replaced. Disruption of knowledge continuity leads to a negative impact on the environment.

**Keywords:** forest economy; employment; employment in forestry; forecasting

## 1. Introduction

Employment in forestry is undergoing a radical transformation. On one hand, the number of employees in forestry work is decreasing, on the other hand there is an increased demand for workers in forests, due to the current bark beetle. In addition, there is a need for more temporary workers for redevelopment work and planting. In our article we are dealing with the problem of a decreasing forestry industry employment rate in the Czech Republic and its impacts. However the issue is even more complicated, as there has been an increase in the extraction of calamity wood corrupted by bark beetles (around 5.5 million m$^3$ per year), which means that after in the wake of a calamity crisis, mining in certain areas will be reduced for some time and will not be able to obtain profits for several years (around 4.5 years). The total volume of mining remains at about 15.5 million m$^3$ per year. This demand for mining means that in the future there will be a decrease in demand for forestry work to only around six people. This will interrupt the transmission of specific knowledge and will result in the loss of

skilled forestry workers. Temporary workers in the forest industry (aliens) will return to their home destinations, resulting in insufficient numbers of young workers in the area trained in forestry and forest mechanization. This is due to low wages in the sector, as shown in the following table (Table 1), as well as relatively difficult and unstable work in forestry [1].

**Table 1.** Employees and wages information for forestry activities (the Czech Republic).

| Indicator | Total | Index 2018/2017 |
|---|---|---|
| Average registered number of employees | 13,386 | 101.9 |
| Average registered number | 13,139 | 101.9 |
| of manual workers | 6,553 | 101.1 |
| Wages without other personnel costs (in thousands of CZK) | 4,288,432 | 106.3 |
| for manual workers | 1,629,778 | 105.9 |
| Average crude monthly wage (employees-per capita) (in CZK) | 26,697 | 104.3 |

If we look at the statistics of the average number of employed individuals in the sector, we see stagnation just above the threshold of 13,000 individuals. Last year, there was a slight increase in the number of forestry workers, after remediation of the wood affected by bark beetles began. Overall, however, it is one of the smallest sectors in terms of the number of persons registered as employees [2].

If we examine the average number of registered employees, the numbers have not changed much in the last five years. This does not correspond to the amount of work related to the remediation of calamity wood and the subsequent additional work. Ideally, 6000 more employees would be needed in Lesy ČR sp.

The average number of employees has been stagnant for the fifth year in a row. In 2018, there was a slight increase in the number of employees to 13,386, which was due to the start of the first stage of bark calamity remediation. In 2019, the number will be increased by a further few hundred employees. Additional staff from foreign countries will be employed for forestry and cultivation. The problem, however, is that after the end of the bark crisis, the volume of random mining will drop, as well as the number of workers. It can be said that if there is incidental mining of about 6 million m$^3$, there will be no need for planned mining for several years. This will lead to a further reduction of the forestry staff in the future [3]. The decline in forestry is not only a problem for the present, but also for the next three to five years because it means undermining any continuity of knowledge and being unable to engage enough workers. Even if we look at the average registered number of employees, which are often more objective, the number of persons employed slightly exceeds 13,000, but the demand for workers is another 4000 to 6000.

It is true that low average wages deter interest in forestry work, as shown in the following table. In 2018 (Table 2), the average crude wage in forestry was CZK 28,858. A few thousand higher-paying jobs (34,105 CZK) were for state enterprises, which represent the Lesy ČR sp area. However, most of these employees were management and technical staff.

**Table 2.** Average crude monthly wage (employees-per capita).

| Year | 2013 | 2014 | 2015 | 2016 | 2017 | 2018 |
|---|---|---|---|---|---|---|
| Forestry in total | 23,628 | 24,559 | 24,900 | 25,602 | 26,697 | 28,858 |
| State | 28,048 | 28,981 | 28,716 | 29,283 | 30,237 | 32,977 |
| Private | 20,968 | 21,735 | 22,463 | 23,360 | 24,502 | 26,223 |
| Municipal | 21,021 | 22,276 | 22,416 | 22,558 | 23,949 | 26,301 |

Table 2 shows that there has been an increase in the average crude nominal wages in the forest sector over the last five years. These are wages that are significantly lower than the average nominal wage (CZK 34,263). As mentioned, employment is increasing, but employment in forestry has been declining for a long time. This is a problem for advanced countries, including the Czech Republic as

a European country, and is a relatively negative trend. Although some authors argue that this trend is due to both the growth of forestry intensification and the use of machinery, we suggest that the decline in the number of forestry workers in the Czech Republic is untenable in the long run. In the following chapters, we therefore look at calculations of the decline in forestry employment and forecasts of this development. We will also try to propose a solution to this problem in terms of economic, social, and environmental sustainability [2].

The aim of the study is to define the development of employment of the number of forestry workers in the Czech Republic. The forecast should show long-term developments from 1930 to 2019. Using the forecast and results of the analysis, a certain strategy for solving the problem of employment in forestry should emerge. The decrease in the average number of registered forestry workers is an economic problem. However, in a wider sense it is also a social and environmental problem.

Employment in forestry affects not only profitability in the sector, which is of interest to forest owners, but also the quality of the environment. For example, bark beetle has impacts on the price of wood, higher logging costs, and thus uncertain profits for forest owners. It also has implications for the wider public. The quality of the environment, tourism, and sustainability of water is decreasing.

Forestry employment is a key issue for us. The solution is not only a short-term increase in the number of employees working in the forest, but also the development of forest sustainability in the future. This includes the retention of key workers in the sector and the maintenance of specific and complex forestry knowledge. Increase of short-term staff does not eliminate this threat of interruption to the continuity of knowledge. The solution is the long-term concept of the development of the sector and the strategy of employee care. This includes the systematic expansion of forestry education.

The purpose of forecast on the development of employment in forestry is that it can be used for managerial and personal decision-making. Forestry businesses, governments, and self-governing organizations can draw on this forecast. The forecast is intended not only for commercial purposes, but also for public interest. It can be used to build personnel and wage policy goals in forestry and can serve as valued material. The objectives of the thesis are therefore not only theoretical because the methodology of data work (time series analysis), but also practical.

## 2. Materials and Methods

Time series means the sequence of values of the selected economic indicator, which is defined in terms of substance and space. We organized values from the past to the present with the possibility of extrapolation to the future. Time series characterize the trend and seasonality, respectively, using nonlinearity and heteroscedasticity. Time series analysis serves to predict future values. There are several approaches to the composition of time series. From the point of view of the long-term trend in forestry employment, it is necessary to choose different means of analysis. The time series are interval and instantaneous. In line with empirical evidence, we use momentum because we have values from 1930 to 2017. Interval indicators are dependent on the length of the interval and it is possible to generate sums. To avoid distortion, the indicators should be related to the same lengthy intervals. The instant indicators refer to a certain time or year. The periodicity of the data was the length of the period to be distinguished on the long-term and short-term time series. The short-term time series refers to a period of less than one year. We have annual employment indicator values available, which is a long-term time series. For long-term (annual) time series, the periodicity duration is expected to be longer than one year.

The time series are displayed so that the graph affects their essential features and properties. The most common time series view is a line graph where a time variable is located on the horizontal axis, and the time series values are plotted on the vertical axis. Other types of graphs include, for example, a box or graph of seasonal values [2].

We will also use some descriptive characteristics and rates of dynamics in the methodological approach. When analyzing time series, it is sometimes essential to calculate the average values of the monitored variable. Using a simple measure of dynamics, the basic slopes of the time series can

be characterized and the criteria for their modeling are identified. We use the following selected descriptive characteristics, such as a simple arithmetic mean:

$$x = \frac{\sum_{j=1}^{n} x_j}{n}.$$ (1)

For the calculation of average values for interval time series, we need a plain arithmetic mean. The values of the time series are marked with $y_t$ the length of the time series is expressed $n$. Plain chronological average is calculated as follows:

$$\bar{y}_{ch} = \frac{1}{n-1} \sum_{t=2}^{n} \frac{y_{t-1} + y_t}{2} = \frac{1}{n-1}\left(\frac{y_1}{2} + y_2 + \cdots + y_{n-1} + \frac{y_n}{2}\right)$$ (2)

A necessary precondition is a time series $y_t$, $t = 1, n$. Another tool for the composition of the time series is the absolute increment:

$$\Delta y_t = y_t - y_{t-1}, t = 2, 3, \ldots, n.$$ (3)

It is easiest to calculate the rate of intensity ($t = 2, n$). The value compares time ($t$) and the change in time ($t - 1$). It is used often, and the average absolute increment provided is as follows:

$$\overline{\Delta} = \frac{\sum \Delta y_d}{n-1} = \frac{(y_2 - y_1) + (y_3 - y_2) + \ldots + (y_n - y_{n-1})}{n-1} = \frac{y_n - y_1}{n-1}.$$ (4)

It is also included in the model by using the first difference of the growth factor. Differentiation (the first, second, and third differential) has a role in mathematic modeling and trend analysis for the selection of time series. In our selection there will be a limited number of trends, from which a valid model will be chosen. The growth factor is:

$$\bar{k} = \sqrt[n-1]{k_1 \, k_2 \ldots k_n} = \sqrt[n-1]{\frac{y_2}{y_1} \frac{y_3}{y_2} \frac{y_4}{y_3} \ldots \frac{y_n}{y_{n-1}}} = \sqrt[n-1]{\frac{y_n}{y_1}}.$$ (5)

Subsequently, the modeling and decomposition of the time series shows the extent to which the forecast is valid and meaningful. The time series is used to form a one-dimensional model where $y_t$. expresses the variable value at time ($t$), the change in time ($t - 1$), and the growth of the forestry sector ($\varepsilon_t$) for the value of the random folder at time ($t$) of the classical method of modeling. It is based on the Box-Jenkins methodology, or on spectral analysis. Followed by decomposition [4], this allows us to interpret the forecast. In the model of economic time series $y_t$ where $t = 1, 2, n$, the components of the trend are $T_t$, seasonal and cyclical ($C_t$), and the random folder ($\varepsilon_t$).

$$y_t = T_t + S_t + C_t + \varepsilon_t.$$ (6)

The trend expresses the long-term vector of the development of the surveyed indicator. It is possible to describe the vector using trending functions (linear, quadratic, exponential, logarithmic, or polynomic) if a series is strongly influenced by a random component or has extreme values. Then, a model of moving averages can be used [5]. Seasonal fluctuations occur during the period of one year. Given that we have annual data, the seasonal layer will be minimal because of the variation of the season. The cyclical component is the fluctuation around the trend where the periods of growth and decline alternate. The periods are an irregular trend factor (they have different lengths and amplitudes) and are formed over a period longer than one year. The formation of cycles can have both an economic and non-economic character. In the econometric model, the random component is a variable that has random fluctuations [5]. This is an important part of the time series analysis; therefore, its properties

must be verified by tests [6]. In the creation of the forecasting model, the decomposition method has a significant impact on validity.

Alternatively, one can use the multiplicative form of decay, where the time series values are generated using the product of the individual components. In both models, the analytical or mechanical alignment methods are used to describe the trend [5] The analytical method can be used to smooth the entire time series at once. It is offset by only one trend function. In the framework of mechanical equalization, we replace empirical values with a range of diameters. The trend folder can be described using trending functions that will be selected (linear, exponential, or logarithmic). In the practical part, fundamental trending functions will be applied, and they usually do not have an asymptote. We describe the individual parameters of trending functions using the smallest squares method, if the function is linear in parameters. The methodology will include the application of the linear trend where there are indications of the presence.

$$y_t = \beta_0 + \beta_1 \cdot t \tag{7}$$

The first difference $y_{t+1} - y_t$ is approximately constant. This means that the trend analysis has result that are monotonous, growing, or, in our case, falling, which determines the sign of the coefficient $\beta_1$. Other parts of the analysis of the trend could still be a quadratic trend:

$$y_t = \beta_0 + \beta_1 t + \beta_2 t^2 + \beta_3 t^3 + \ldots + \beta_S t^S. \tag{8}$$

The trend analysis should yield an estimate with the highest possible validity. Specifying parameters can best be applied as an expression for the OLS to minimize the criterion from the default regression specification:

$$Min \sum_{t=1}^{n} [y_t - \hat{y}_t]^2 = Min \sum_{t=1}^{n} \left[ y_t - \sum_{j=1}^{2} \beta_j f_j(t) \right]^2. \tag{9}$$

Here it is entered as:

$$Min \sum_{t=1}^{n} \left[ y_t - \hat{y}_t \right]^2 = Min \sum_{t=1}^{n} [y_t - \beta_0 - \beta_1 \cdot t]^2. \tag{10}$$

By applying the standard procedure of derivation in both unknown parameters, we get a system of two necessary conditions of extremes:

$$2 \sum_{t=1}^{n} [y_t - \beta_0 - \beta_1 \cdot t](-t) = 0. \tag{11}$$

respectively.

$$2 \sum_{t=1}^{n} [y_t - \beta_0 - \beta_1 \cdot t](-t) = 0. \tag{12}$$

After editing:

$$\sum_{t=1}^{n} [y_t - \beta_0 - \beta_1 \cdot t] = 0, \tag{13}$$

respectively.

$$\sum_{t=1}^{n} [y_t - \beta_0 - \beta_1 \cdot t] = t = 0. \tag{14}$$

$$\sum_{t=1}^{n} y_t - \beta_0 \cdot n - \beta_1 \sum_{t=1}^{n} t = 0, \tag{15}$$

respectively.

$$\sum_{t=1}^{n} y_t - \beta_0 \cdot \sum_{t=1}^{n} t - \beta_1 \sum_{t=1}^{n} t^2 = 0 \tag{16}$$

After totaling expressions:

$$\sum_{t=1}^{n} t = \frac{n(n+1)}{2}, \ \sum_{t=1}^{n} t^2 = \frac{n(n+1)(2n+1)}{6}. \tag{17}$$

And by dividing the numerator and denominator we get the parameter $b_1$:

$$b_1 = \frac{\sum_{t=1}^{n} ty_t - \frac{(n+1)}{2} \cdot \sum_{t=1}^{n} y_t}{\frac{n(n+1)(2n++)}{6} - \frac{n(n+1)^2}{4}} = \frac{\sum_{t=1}^{n} ty_t - \frac{(n+1)}{2} \cdot \sum_{t=1}^{n} y_t}{\frac{n(n+1)[4n+2-3(n+1)]}{12}} = \frac{\sum_{t=1}^{n} ty_t - \frac{(n+1)}{2} \cdot \sum_{t=1}^{n} y_t}{\frac{n(n^2-1)}{12}}. \tag{18}$$

And for the second parameter $b_0$:

$$b_0 = \overline{y} - b_1 \cdot \overline{t} = \overline{y} - \frac{n+1}{2} b_1. \tag{19}$$

The purpose of trend analysis is to estimate the evolution of employment in the forestry sector. We used this to find the optimal type of trend. We utilized expected trends and properties that resulted from the previous analysis. The prognosis is expressed graphically.

## 3. Results

The results of the calculation show that the development of the number of employed in the forestry and logging sector has been decreasing for a long time. There is only a limited time series in our calculations, so the calculation may not be completely accurate. However, it is clear that while the number of employees was 31,800 at the beginning of the period under review, in the following years there was a decrease by a thousand persons each year. It is likely that in the year 2020, around 22,000 people will be employed in the industry. Future developments will depend mainly on investment in the industry, as more forestry investments and newer mining and transport technologies mean the number of jobs will continue to be reduced. As shown in the following graph, there is a weak dependence between foreign direct investment (FDI) and the number of jobs. It should be added, however, that FDI statistically appears as a sum of real and financial investments. Conversely, real investment is related to the purchase of new technologies (mining), while employment, as already mentioned, is trending downwards. The graphs, showing the general trend of reduced forestry employment, point to a general trend of reduced forestry employment in almost all of developed Europe. In the EU-28, there has been a significant reduction in employment over the last five to ten years, ranging from −6.2% in Turkey to −90% in Albania (albeit this reduction was very marginal in that country). Even more critical is the fact that employment in countries with a large pulp and paper industry (more than 10,000 employees) has also reduced significantly: Poland (−38%), Romania (−35%), Ukraine (−31%), Czech Republic (−19%), Portugal (−18%), Slovakia (−17%), Norway (−17%), Germany (−15%), Finland (−15%), and France (−14%).

Calculations using a linear trend analysis show a dynamic drop in the indicator, which corresponds to an empirical point of view. The decline in employment in forestry has been sharply decreasing since 2000. A linear trend analysis shows the rate of decline. $R^2$ is calculated at 0.7771, meaning 78% of the linear model can be explained. This is not the highest validity of all selected models, but does show the degree of decline in the number of employees in forestry, so the model is reliable (Figure 1).

Before we arrive at the valid model (Figure 2), we have to test the logarithmic pattern that is in the graph (Figure 3). Trend analysis shows that 77% of model variables can be explained, which is a relatively good result for prediction. The logarithmic trend also shows a sharp decline in forestry

workers, but it does not include trending and subtracted values in the model, although they are important for the time series (Figure 3).

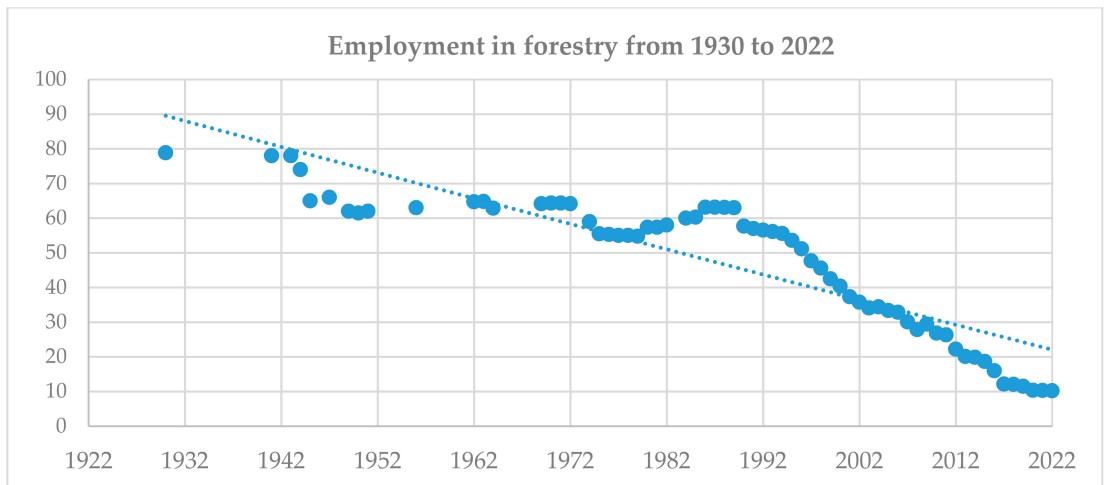

**Figure 1.** A linear trend was obtained through the following equation: (y =−0.0134$x^2$ + 52,344x − 51,012) along with the index of determination ($R^2$ = 0.7771).

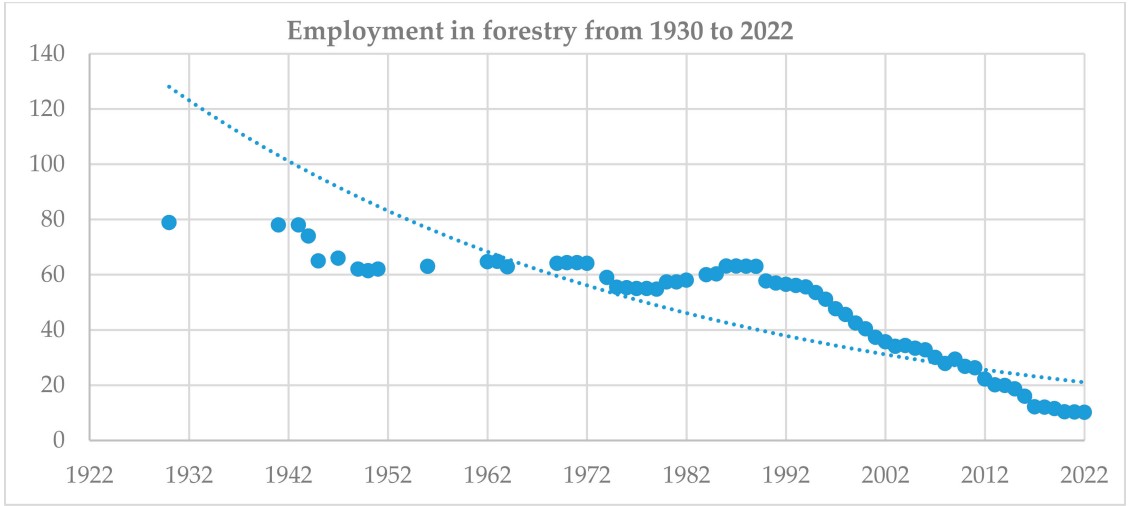

**Figure 2.** This exponential trend was obtained through the following equation:(y = 4E + 134$x^{-40.33}$) along with the index of determination ($R^2$ = 0.5974).

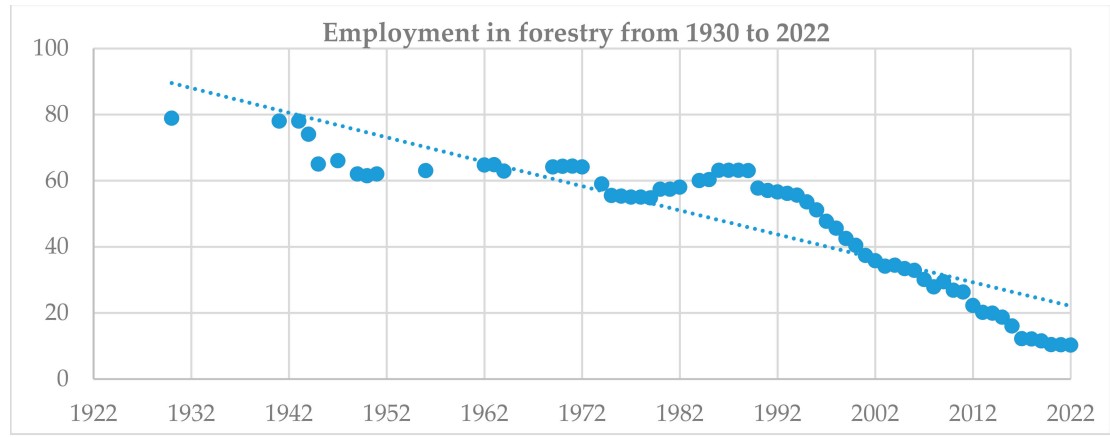

**Figure 3.** This logarithmic trend was obtained through the following equation: ((y = −1473ln(x) + 11,232) along with the index of determination ($R^2$ = 0.7729).

The results of the polynomial trend analysis show a gradual decline in forestry employment. The trend is valid in this case, as nearly 93% of the variables can be explained by the polynomial trend model. This means that only one out of ten predictions will fail. However, the outcome of the trend analysis can still be considered as critical from the point of view of the lack of forestry workers and the lack of graduates of forestry schools. The trend points to the fact that a shortage of workers and a drop in employment in forestry can seriously disrupt forest protection and forest care. Incorrect interventions or late interventions in the field of discharging calamity wood can result in relatively high economic and environmental damage. The practice of removing wood when the bark beetle is transported across the Czech Republic shows that mistaken management can make this situation worse (Figure 4).

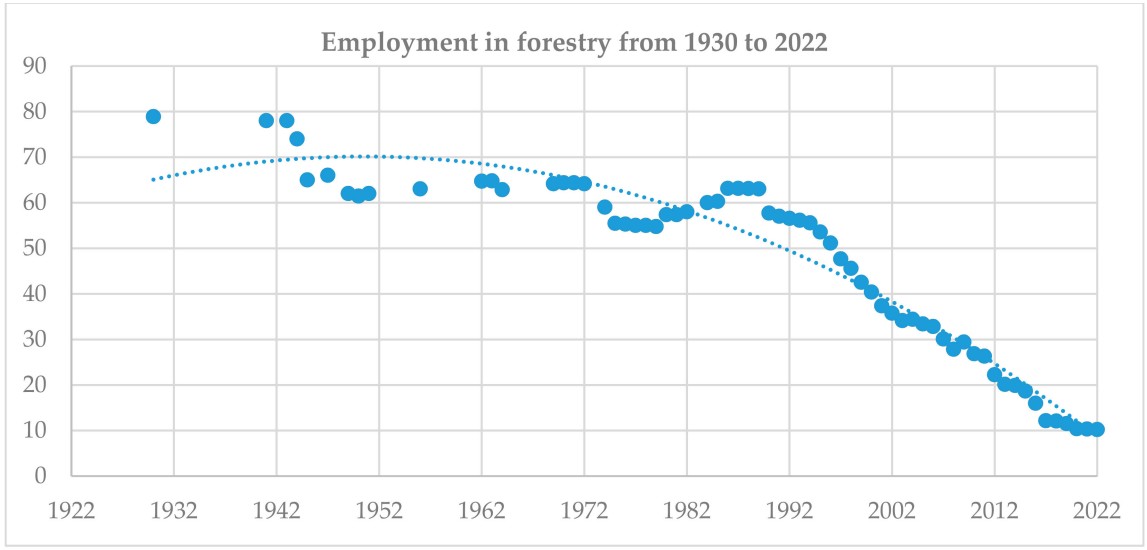

**Figure 4.** This polynomial trend was obtained through the following equation: ($y = -0.0134x^2 + 52,344x - 51,012$) along with the index of determination ($R^2 = 0.9261$).

The previous model, constructed by moving averages, can predict employment decline for one or two future periods. The model shows a fall in forestry employment well below 10 thousand forestry employees. The following table shows more accurate numbers. There is significant risk in the year 2020 when it is clear that the number of people employed in the forestry industry will be only 9623. This is, in terms of comparison with other countries, the lowest number of forestry workers per 10 million inhabitants (Figure 5).

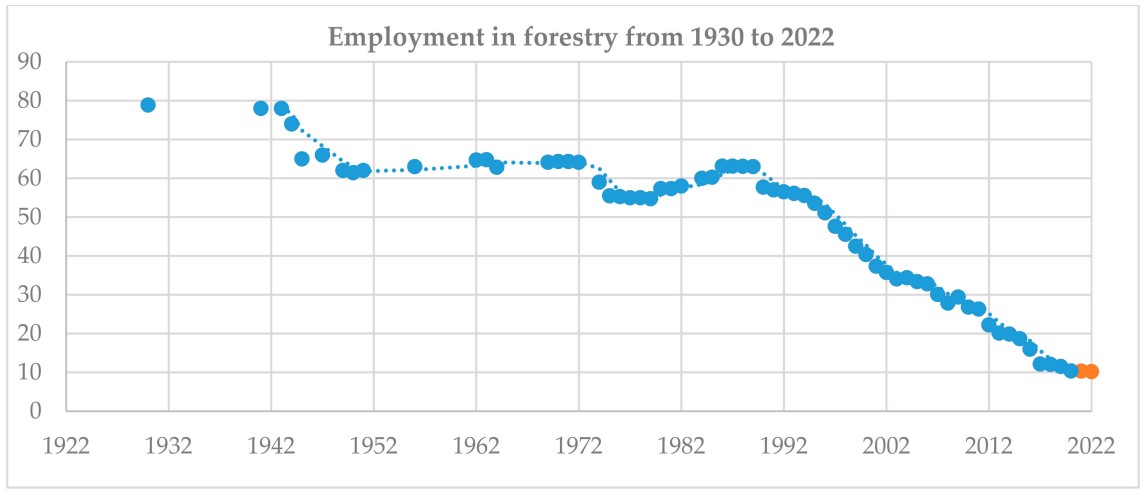

**Figure 5.** A time series alignment.

The model shows that 78% of the variables can be explained by the model. In terms of employment trends, this is a negative trend, as calculations show that a reduction in employment in 2020 could jeopardize the sector's activities and seriously disrupt the ecological and economic balance of the forest, particularly in the case of calamities (Table 3).

**Table 3.** Employment Calculation–Compensation.

| Years | Number of Employees $y_i$ | $t_i$ | $t_i^2$ | $y_i t_i$ | Equalized $Y = a + b t_i$ |
|---|---|---|---|---|---|
| 2006 | 30,058 | 5.5 | 30.25 | −165,319 | 31,266 |
| 2007 | 2783 | 4.5 | 20.25 | −125,235 | 29,553 |
| 2008 | 29,405 | 3.5 | 12.25 | −102,917.5 | 27,839 |
| 2009 | 26,831 | 2.5 | 6.25 | −67,077.5 | 26,125 |
| 2010 | 26,306 | 1.5 | 2.25 | −39,459 | 24,412 |
| 2011 | 20,123 | 0.5 | 0.25 | −11,115 | 22,698 |
| 2012 | 13,043 | 0.5 | 0.25 | 10,056 | 20,984 |
| 2013 | 13,053 | 1.5 | 2.25 | 29,820 | 19,271 |
| 2014 | 12,863 | 2.5 | 6.25 | 46,635 | 17,557 |
| 2015 | 12,853 | 3.5 | 12.25 | 55,954.5 | 15,843 |
| 2016 | 13,139 | 4.5 | 20.25 | 57,600 | 14,130 |
| 2017 | 12,058 | 5.5 | 30.25 | 66,005.5 | 12,416 |
| 2018 | 11,808 | 5.6 | 31.88 | 73,600.5 | 11,808 |
| 2019 | 10,369 | 5.8 | 32.22 | 77,232 | 10,555 |
| 2020 | 10,366 | 6.1 | 33.32 | 78,112 | 10,414 |
| 2021 | 10,311 | 6.0 | 33.41 | 79,661 | 10,355 |
| 2022 | 10,215 | 6.2 | 33.51 | 80,511 | 10,251 |

Sharp drops in employment in the Czech Republic (Czechoslovakia) have existed in the past. The first sharp decline occurred in the 1940s and 1950s. This was due to industrialization and urbanization (labor migration). Because of the mechanization of forestry, the number of workers (in the Czech lands) dropped from 80,000 to 60,000 in less than two decades. By the end of the 1980s, however, the number of workers had stabilized. Another sharp drop in registered forestry workers occurred in the 1990s and has continued since then, although it has stopped in the last year. It is not our aim to monitor the historical context of employment in forestry, but it is important to observe both the long-term trend (from the 1930s) and the medium-term trend (from the 1990s) to the decomposition of the trend (Figure 6).

The Interval Forecast, which is graphically illustrated in the previous figure, shows possible scenarios of development. A realistic scenario (the green part of the chart, the forecast center) shows the employment development forecast as a slight decrease below the ten thousands of registered employees. The pessimistic scenario shows a decrease in the registered number of employees in the sector to 6000 workers. The optimistic scenario (upper limit) shows a slight increase in the number of employees per 15,300 people, which is about 3000 more than the current state. It has to be stated objectively that, in reversing the year-on-year decline in working in the forest, any potential increase will be added to workers in technical and management positions.

Using interpolation criteria, we can compare each model (Figure 6). The table shows the results of the interval forecast (Table 4). As the most appropriate model, a nomic model was chosen with an R value (as the confidence value) of 0.9261. Based on this model, which explains more than 92% of the variables, the future development of the time series can be predicted (Table 5).

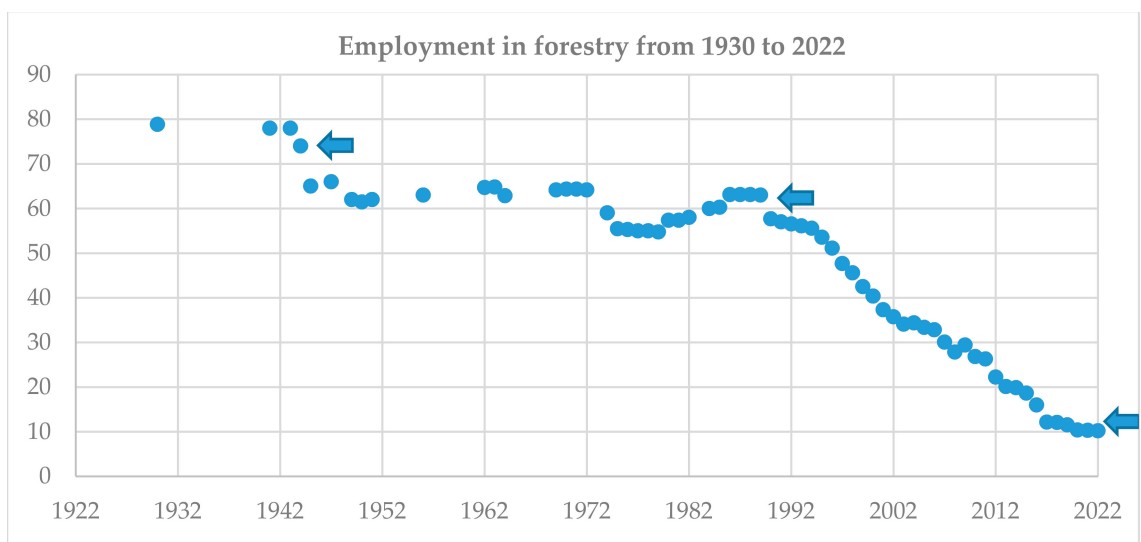

**Figure 6.** Prognosis of employment in forestry-time breaks.

**Table 4.** Interval forecast of employment in forestry by 2020.

| Years | Employees | Minimal | Middle | Maximal |
|---|---|---|---|---|
| 2011 | 26,831 | | | |
| 2012 | 26,306 | | | |
| 2013 | 22,230 | | | |
| 2014 | 20,112 | | | |
| 2015 | 19,888 | | | |
| 2016 | 18,654 | | | |
| 2017 | 15,987 | | | |
| 2018 | 12,800 | | | |
| 2019 | 11,055 | | | |
| 2020 | | 9501 | 11,500 | 13,000 |
| 2021 | | 9450 | 10,250 | 11,520 |
| 2022 | | 9350 | 10,001 | 11,250 |

**Table 5.** Model Comparison.

| | Trivial | Proportional | Constant |
|---|---|---|---|
| M. E. | 0.92 | 0.06 | 0.00 |
| M. A. E. | 349.08 | 350.08 | 354.93 |
| M.A. P.E. | 162,682.97 | 159,401.07 | 163,919.03 |
| M. S. E. | 403.34 | 399.25 | 404.87 |

The model of testing errors displays very low values. Therefore, the used prediction model has high validity (Table 5). In the study, we have indicated a continuing downward trend in relation to amount of people employed in forestry, as well as a decrease in the employment rate. The forestry industry in the Czech Republic is currently facing a human resources crisis. The lack of skilled labor is apparent in all forestry professions. The lack of workers is to some extent compensated for by temporary foreign workers, but the problem still remains. For the next decade, it will be necessary to provide human resources for forest regeneration, along with for the maintenance and management of forestlands (which are being affected by a bark beetle calamity and are therefore are being felled in large numbers).

## 4. Discussion

The results of the calculation show that the number of employees in the forestry and logging sectors has been declining for a long time. In our calculations, there is only a short time series, so the calculation may not be accurate. However, if 31,800 people were employed at the beginning of the reference period, then a decrease of 1000 people per year occurred in the following years.

The aim of the study was to analyze the possible development of employment in the forestry sector based on a time series analysis. The target was defined so that the development of employment in forestry by the end of 2020 can be interpreted from the results. The decline in the number of employed is critical. While the Forestry Statistical Indicator does not include all forest work, the number of people expected to be employed is below 10,000 in the coming years, which is a critical threshold for any sector of the economy [2,7]. The results also correspond to the previous predictions of forestry development [8] and its structure [9], even though the demands on the use of non-productive forest functions are increasing [10].

The impact of forestry on employment in the Czech Republic has a broader economic significance than the statistics show [10]. The values of employment in forestry are around 0.6% to 0.7%, which is a statistically low proportion [11]. However, we need to also consider the technological development of efficient mining, the use of modern harvesters, transport technology, and IT [12] "transferring" jobs from forestry to broader industry, whose share of economic indicators is growing [13]. Another very important factor is the economic importance of environmentalism [14]. Current lifestyles create a demand for natural materials in construction, residential architecture, and other areas [12]. The consumer then purchases more wood products, more construction projects, and buildings that use wood. All of this increases both job creation (and employment) and other economic indicators [13], which are statistically reported as timber, furniture, and paper industries [8]. This is a multiplier effect of forestry [15]. At present, the usage of wood is a global phenomenon.

These trends do not only affect economically developed countries, but also emerging economies [16]. Consumers of these countries are eager for bio or green goods and products, with natural wood materials coming to the fore. They can replace plastics or synthetic materials worldwide. While wood demand is declining in the short term, it will continue to rise in the long term [13]. This is related to the growth in the number of jobs [17], which we consider statistically to be in the manufacturing or construction industries but are derived from forestry [13]. Thus, the importance of forestry and forestry can be seen in both economic (production) and non-production levels (care for the landscape, water, and air protection), but also in the social and educational levels [10]. Employment as a macroeconomic phenomenon is highly dependent on investment. It is proven that the employment rate and the number of jobs are mainly influenced by the so-called real investment (not including the financial investment that has already occurred) [16]. This is because new jobs [13] are created by investing in new technologies, production facilities, processes, etc.

Wages are also growing in forestry, and are currently 28,858 CZK [18–21]. The growth rate of wages in forestry is not very high, but the trend is rising in the long run. The time series of foreign direct investment shows that the trend in foreign investment growth is growing, which means that employment in the forestry sector is likely to fall. This is because as investment in forestry and timber production increases, the number of sophisticated mining processes and new technologies (harvesters) can also be expected to grow, causing a steady decline in job creation [8], especially in manual labor [15]. Although there is some social deprivation for employees over the age of 50, their presence in forestry is growing somewhat [22]. The new jobs that will be created in forestry and forestry will surely require higher qualifications (requirements for operators of mining machinery, monitoring, recording and tracking devices, demands for knowledge of biotechnology and biotechnology, and demands for knowledge of information and communication technologies are increasing) and more experience [23]. Whether for mining machinery, harvesters, transport technology, or for woodworking workers, there will be an increasing emphasis on qualification levels, which is a challenge for secondary and

higher education systems [24]. Forestry education and related disciplines such as environmental economics and forest management are of great importance [24].

## 5. Conclusions

For our model, we can exclude a seasonality component that does not affect it. Non-cyclical effects, such as calamities, remain a determinant. Over time they are repeated irregularly and can cause residual fluctuations. This corresponds to the current situation in forestry, where the bark beetle calamity requires more than 4000 new employees for redevelopment and subsequent reclamation work. Whether these are employees of the Czech labor market or foreign nationals cannot be inferred. Employment development scenarios result from calculations using the model. A confidence interval of 95% was set for the calculation. However, the range would be 70%. The calculated p-value for 2019 is 0.1658, and the p-value for 2020 is 0.2474. This means that the values are higher than 0.05, so we can assert that the forecast is reliable.

The empirical results have shown that the shortage of forestry workers may have an impact on environmental protection. The findings are extremely important in terms of environmental policy-making in the Czech Republic. Economic investments are needed to make the transition to cleaner technologies but a skilled labor supply in the sector is also mandatory. Political and legislative measures to support job creation in relation to environmental activities are also needed to achieve the fully sustainable success of forestry businesses. New jobs that are emerging in forestry place higher demands on employees in terms of education and technology practices. Universities and schools are also facing these demands. This is also a political challenge. The state environmental policy of the Czech Republic must include support for so-called new and sustainable jobs.

The decreasing number of employees in the forestry sector along with an additional loss of skilled workers involved in forestry is causing serious problems. Forestry experts are aware of this issue. Therefore, the Active Employment Policy reflects the newly emerging State Environmental Policy. The new Environmental Policy should include supportive socio-economic tools that will lead to the creation of new "Green Jobs". Our empirical results show that the State Environmental Policy needs further development, because socio-economic aspects are a very important part of environmental protection and nature sustainability. Our prediction implies that it is necessary to support forestry employment and provide support for environmental protection strategically and in the long term. The results of our research will be mainly important for governmental experts involved in the development of the State Environmental Policy for the years 2020 to 2030. According to the results of this study, there is an option of using future economic tools that should be politically and consensually accepted.

**Author Contributions:** Conceptualization, D.T.; Data curation, D.T.; Formal analysis, K.M.; Funding acquisition, M.M.; Investigation, M.M.; Methodology, D.T.; Resources, M.M.; Supervision, K.M. and D.T.; Validation, M.M.; Writing, M.M.; review & editing, D.T. and M.M.

**Funding:** This research received no external funding.

**Acknowledgments:** This paper was supported by the Internal grant agency (IGA) of the Faculty of Economics and Management, Czech University of Life Sciences Prague, grant no. 2019B0011 "Economic analysis of water balance of the current agricultural commodities production mix in the Czech Republic" (Ekonomická analýza vodní bilance stávajícího produkčního mixu zemědělských komodit v ČR).

**Conflicts of Interest:** The authors declare no conflict of interest.

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
