# Peer review of "Development and Forecast of Employment in Forestry in the Czech Republic"

_sustainability, doi:10.3390/su11246901_

Round 1

Reviewer 1 Report

The development and forecast of employment in forestry in the Czech Republic is really a good work based on some assumptions. However, some revision will provide the information in more constructive way.

Abstract has mainly focused on background information rather the results of the study; better to focus more on the study findings. Finally, the main limitation is that the forecast is only explaining in the EU context. In introduction, the background information must be reflected in wider aspect rather only focusing on CZK perspectives; I mean lack of worldwide background. It’s better to focus more on world forestry employment information rather mining example in the beginning. Table 1 and its items are somewhat complex to understand; Table 1 & 2 missing the sources. At the end of the problem statement the authors articulated many broad issues; better to focus on specific issue and then draw the objective. In Materials and Methods, the heading and sub-heading are confusing and misleading the reader. For example, after study area, may be Analytical Frameworks should be best fitted heading to describe the calculation/equations. I am wonder why authors discuss the calculation in methods section. Better to merge the main description with the calculation, and also placed basic discussion after the Results section with a separate heading of Discussion section. Need to separate the results with discussion, this section only focusses the result only not discussion/elaboration. Try to cite reference from other part of the world; not only focused on EU context, it will reduce the merit of the study. Table 4 is not clearly understandable and the abbreviation should be avoided from Table 5. In Conclusion, the future directions are missing; try to formulate the concluding remarks with a future direction for the readers based on study’s findings. Most of the references are from CZK context, try to explore it with world perspectives.

Good Luck

Author Response

All the following changes are in regime „track changes“. Abstract has been updated to provide also the results of the study The content of our manuscript bases on EU context & Czech background not worldwide. The chapter regarding world context of forestry has been included to part of „Discussion“. Table 1 along its items was simplified and the source has been included for this table along with table 2. The statement in the article was updated, more appropriate form where we as authors focus on specific issue. The headings and subheadings in section Materials and Methods are no longer confusing for readers. We updated the section „discussion“& „methods“. The section “results” now displays separately. So does section “discussion”. Table 4 has been edited and the values were added to match prediction for year 2022 Table 5 – the data were re-calculated and updated. The Conclusion of the manuscript is now more information and the up-to-date data were included.

Reviewer 2 Report

This study employs the basic econometric model to analyze the trend of employment in the forestry sector based on a time series analysis. However, no sensitivity analysis for parameters. Or how to prove the accuracy of the simulation instead of R-square? Besides, there are several problems need to be addressed such as the model for prediction, the results/conclusion such as the multiplier effect, and many mistakes on writing, etc.

In this study, authors want to make a long-term prediction for the employment trend. However, prediction to 2020 is not long enough in terms of long-term especially it is the end of 2019. I don’t think it has a forecast on the employment for the long-term of the Czech Republic as described (two-year prediction cannot be called long run). In the figures of results. Are lines stop at 2020 but the legend is to 2030/2040, then why not predict to 2030/2040 to make a long-run prediction?

Study periods are not consistent in the manuscript. “The objective is to define the development of employment of the number of forestry 84 recorders in the Czech Republic. The forecast should show long-term developments from 1930 to 2017.” But all results are for 1930-2020. Descriptions are consistent. Please double check and keep consistent.

This study analyzes the trend of employment in the forest sector using 84 recorders in the Czech Republic as employment in the forest sector benefits both economic and environment. However, no results to prove the employment benefits on the environment and economics.

In this manuscript, authors talk about the multiplier effect twice: one is in the abstract as a highlight for this research; another is in the discussion (Page 7 line 216: “However, it is a multiplier effect of forestry.”). There is no statistically jurisdiction/proof on the multiplier effect. How do you prove/justify multiplier effect in your research rather than using definition to explain. And what’s the magnitude of this effect?

In addition, more attentions should be concentrated on the following parts:

In the abstract, 14 proportion of employees in the forestry sector is relatively small standing at only 0.6%. This is for the Czech Republic or the world?

Table 1 shows the employees and wages in forestry activities. It’s for the whole world or the Czech Republic?

Page 2 line 77: Even in our country, employment in forestry is declining. What does our mean?

Page 13 line 341: Based on the model, you can eliminate the seasonality component that does not affect the model. What does “you” mean?

Page 3 line 84 and 104: The aim of the thesis is to define the development of employment of the number of forestry 84 recorders in the Czech Republic. The objectives of the thesis are therefore not only 104 theoretical, because the methodology of data work (time series analysis), but also practical. Thesis?

Page 7 line 193: 2.2 discussion should stay after results.

Page 7 line 198: The aim and aim of the study were to analyze the possible development of employment in the forestry sector based on a time series analysis. Two aims in one sentence. Please double check.

Page 7 line 212: The consumer then purchases more wood products, more construction projects and wood buildings use wood. Wood buildings and use wood have the same meaning.

Page 7 line 217: The trend of using wood and wood is currently a global 217 phenomenon. Two wood in this sentence. There are many this kind of mistakes in this manuscript. Please double check.

All figures have two capitals, it is better to remove the one on the figure.

Author Response

All the following changes are in regime „track changes“. The prediction now meets required criteria for “prediction”. Now displays data till year 2022. We revised entire content of our manuscript. The data for prediction were re-calculated and updated. Graph that displayed prediction for period 2040 is now up to year 2022. All the results are now up-to-date so they are no longer confusing the reader. Therefore, the prediction is now long term. In chapter: conclusion we added “results that prove employments benefits on the environment and economics We erased duplicate mention of word “multiplication”. Those 14% mentioned in abstract are meant only for Czech Republic and this information in text is now clear, and gives reader correct information. Table 1 – now only specializes on Czech Republic. Page 2 line 77: word “our country” changed to “Czech republic”. Page 13 line 341: We grammatically corrected the sentence at this location in manuscript. Page 3 line 84 and 104: Went under revision and there was correction made in the matter. Page 7 line 193: The chapters “results” swapped with “discussion”. Page 7 line 198: We did several grammatical changes in the text. Page 7 line 212: We clarified and made grammatical corrections in this area. Page 7 line 217: We revised this part to sound appropriately. All the Figures now display the title only once. Below each Figure now appears only brief description for the reader.

Reviewer 3 Report

The paper "Development and forecast of employment in forestry in the Czech Republic" is interesting for journal but the current version needs major revisions before further consideration.

The aim of the analysis should be evidenced in the abstract and introduction sections. It should be assessed uniformly through the paper. Moreover, the authors should start with a clear question(s) that will be answered. The objectives and /or research questions section would help to summarize and focus the overall aim of the study and improve the conclusions section, once the main ideas are clear and systematized.

The literature review section is very poor. The following papers could be a guide for an opportune literature review section:

Aldieri L. & Vinci C. P. (2018). Green Economy and Sustainable Development: The Economic Impact of Innovation on Employment. Sustainability, 10, 3541. Van Roy V., Vértesy D. & Vivarelli M. (2018). Technology and employment: Mass unemployment or job creation? Empirical evidence from European patenting firms. Research Policy, https://doi.org(10.1016/j.respol.2018.06.008.

As far as the methodological approach is concerning, I would like to see more robustness checks to determine how sensitive these results are relative to the chosen specification.

The results of the analysis should be further discussed and improved. The contribution can be made evident only putting the accent on the gap in the literature.

The conclusions are too short and should be discussed also in terms of policy implications relative to Czech Republic country.

Author Response

All the following changes are in regime „track changes“. We introduced results of the calculations in the abstract part. We revised the chapter “conclusion”, added with detailed information. We included more sources in the Literature section – The mentioned sources were included. We defined and improved the results and calculation. The whole prognosis is up to date up to year 2020. We revised the “conclusion” and now mentions political implications in the Czech Republic.

Round 2

Reviewer 2 Report

Authors have revised based on comments from the first review. Some revisions needed:

Figure 1 title should be consistent with others, which is 1930-2022;

Page 3 Line 91: "The forecast should show long-term developments from 1930 to 91 2017." Forecast is conducted to 2022 with the data from 1930-2017, right? Please double check.

Author Response

All the following changes are in regime „track changes“.

Figure 1: the title is now consistent with other figures.

Page 3 Line 91: "The forecast should show long-term developments from 1930 to 91 2017." This sentence has been corrected.

The conclusion is no longer short and is discussed also in terms of
policy implications relative to Czech Republic country. The results of the analysis were further discussed and improved.

Regards

Reviewer 3 Report

The paper has been structurally improved. Now, it can be accepted for publication. 

Author Response

Thank you very much for your positive response.

We respect your final opinion.

Regards